# Effects of Group Exercise Intervention on Quality of Life and Physical Parameters in Patients with Childhood Cancer: A Systematic Review

Yurina Doi [1], Masato Ogawa [2,3,4], Kodai Ishihara [3,4,5], Junichiro Inoue [6] and Kazuhiro P. Izawa [3,4,*]

[1] Department of Health Science, Faculty of Medicine, Kobe University, 7-10-2 Tomogaoka, Kobe 654-0142, Japan; 2023919m@stu.kobe-u.ac.jp

[2] Department of Rehabilitation, Faculty of Health Sciences, Osaka Health Sciences University, 1-9-27 Temma, Osaka 530-0043, Japan; mogawa@med.kobe-u.ac.jp

[3] Department of Public Health, Graduate School of Health Sciences, Kobe University, 7-10-2 Tomogaoka, Kobe 654-0142, Japan; mhe1601@std.huhs.ac.jp

[4] Cardiovascular stroke Renal Project (CRP), 7-10-2 Tomogaoka, Kobe 654-0142, Japan

[5] Department of Physical Therapy, Faculty of Nursing and Rehabilitation, Konan Women's University, 6-2-23 Morikitamachi, Kobe 658-0001, Japan

[6] Division of Rehabilitation Medicine, Kobe University Hospital International Clinical Cancer Research Center, 1-5-1 Minatojima Minamimachi, Kobe 650-0047, Japan; jinoue@panda.kobe-u.ac.jp

[*] Correspondence: izawapk@harbor.kobe-u.ac.jp; Tel.: +81-78-796-4566

**Abstract:** Background: Although the survival rates of childhood cancer are increasing, children diagnosed as having cancer experience psychological and physical problems and a declining quality of life (QOL). Methods: A systematic review of PubMed databases was conducted up to September 2023 to identify studies reporting the effects of group exercise intervention in children with cancer. The inclusion criteria were pre-specified, including children aged ≤19 years old who received group exercise intervention and interventional studies written in English. Studies involving non-exercise intervention or non-group intervention were excluded. Results: Five studies were included in the present review. In three studies, QOL and physical parameters were improved after group exercise intervention, and in two studies, only physical parameters were improved. Improvements in QOL were achieved through psychosocial variables, improved scores of subscales of pain and hurt, nausea, and procedure-related anxiety, and reduced cancer-related fatigue. All studies had high numbers of participants who completed the intervention. However, all studies showed a high risk of bias regarding the selection of the reported results, and most studies showed a high risk of bias regarding deviations from the intended intervention and outcome measurement. Conclusion: The reviewed studies showed that group exercise intervention for children with cancer could improve their QOL and/or physical parameters.

**Keywords:** childhood cancer; group intervention; exercise; quality of life; physical parameter; systematic review

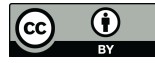

## 1. Introduction

Survival rates for childhood cancer are increasing due to advances in childhood cancer care, with high-income nations having achieved notable advancements that are yielding survival rates exceeding 80% [1]. However, the psychological and physical repercussions for afflicted children remain profound. Children diagnosed as having cancer experience psychological problems such as lack of self-esteem, emotional regression, and depression [2]. Moreover, they also experience physical problems such as muscle weakness [3,4], poor physical or cardiorespiratory function [5,6], and low physical activity [6,7]. These problems can lead to a decline in their quality of life (QOL) [8–10].

Studies focusing on exercise intervention in patients and survivors of childhood cancer have indicated that physical activity can enhance their QOL. In these patients, exercise intervention improved the parents' evaluation of their children's QOL for the general domains and for the fatigue scale, although the children themselves self-reported no changes in their QOL [11]. Exercise in childhood cancer survivors had a positive effect on their QOL regarding physical functioning, general health, vitality, emotional role limitations, and mental health domains [12]. However, several studies have reported high dropout rates from exercise training in patients and survivors of childhood cancer. This may be due to the difficulty in combining exercise with their other activities, program boredom, perceived lack of need for exercise, and physical problems [13,14].

As one answer, exercise interventions that include family/peer interaction and a combined program of exercise and play have been reported to partially contribute to improvements in participation rates in exercise intervention, physical and psychological aspects, and QOL in patients and survivors of childhood cancer [15–17]. However, there remains no consensus, and opinions vary widely among researchers.

Therefore, the purpose of this study was to clarify the relationship between group exercise training and QOL and physical parameters in patients and survivors of childhood cancer via a systematic review of published studies.

## 2. Materials and Methods

### 2.1. Eligibility Criteria

This systematic review was conducted based on the Preferred Reporting Items for Systematic Reviews and Meta-Analyses (PRISMA) statement [18]. Inclusion criteria were patients with childhood cancer, patients aged ≤19 years, intervention with exercise training, only published studies written in English, publication date from January 2000 to September 2023, and interventional study. Exclusion criteria were intervention without exercise, not group intervention training, observational study, review, meta-analysis, study protocol, and editorial. This study required no ethical approval as it was based solely on published literature.

### 2.2. Search Strategy

Studies were searched in the MEDLINE (PubMed) database only. The search was conducted on 27 September 2023. Keywords related to "childhood cancer", "children", "exercise", "group therapy", and "clinical trials" were used (Supplementary Table S1). A manual search was also conducted.

### 2.3. Selection Process

The selection process consisted of first screening, second screening, and manual research. Before the first screening, 5 articles from other sources were added. In the first screening, the titles and abstracts of each study were read to check whether the article met the inclusion criteria. In the second screening, the full text of the articles included was read, and those meeting the exclusion criteria were excluded. This process was conducted independently by two reviewers (Y.D. and K.I. or M.O.), and disagreements about screening results not resolved by consensus were discussed with a third reviewer (K.I. or M.O.). Author Y.D. transferred data from the data extraction form to the Review Manager file. Then, the Rayyan software (Rayyan—AI Powered Tool for Systematic Literature Reviews, https://help.rayyan.ai/hc/en-us, accessed on 27 September 2023), a recommended screening system, was used to manage literature records and data.

### 2.4. Data Collection Process

The following information was extracted from the articles included: number of participants, patients' characteristics (age/diagnosis), intervention (type of exercise, duration, and follow-up), group intervention elements, outcomes, and main findings.

*2.5. Risk of Bias Assessment*

Two researchers (Y.D. and K.I. or M.O.) independently assessed the risk of bias of the articles included using the Risk of Bias version 2 tool [19], and disagreements about assessment results not resolved by consensus were discussed with the third reviewer (K.I. or M.O.).

Risk of bias was evaluated for the following five domains: the randomisation process, deviations from the intended interventions, missing outcome data, measurement of the outcome, and selection of the reported result. The domain of the randomisation process was evaluated based on the assessment of random sequence generation, concealment of allocation sequence, and baseline imbalance and analyses that adjust for baseline imbalances [19]. The domain of deviations from intended interventions was evaluated on whether participants, carers, and people delivering the interventions were unaware of intervention groups during the trial and whether an appropriate analysis was used to estimate the effect of assignment to intervention [19]. The domain of missing outcome data was evaluated on whether outcome data were available for all, or nearly all, randomised participants, and whether the result was not biased by missing outcome data and that missing outcome could not depend on their true value [19]. The domain of measurement of the outcome was evaluated on the appropriateness of the outcome measurement, whether the measurement or ascertainment of the outcome did not differ between intervention groups, that the outcome assessors were aware of the intervention received by the study participants, and whether the assessment of the outcome was influenced by knowledge of the intervention received [19]. The domain of selection of the reported results was evaluated on whether the data were analysed in accordance with a pre-specified plan and whether the result was selected from multiple eligible outcome measurements within the outcome domain and multiple eligible analyses [19]. These five domains were ultimately adjudicated using an algorithm to determine the risk of bias and reported as "low risk", "unclear risk", or "high risk" [19].

**3. Results**

*3.1. Study Selection*

We obtained 377 relevant articles through PubMed and 5 additional articles identified through other sources. After the first screening, 339 articles were excluded based on the inclusion and exclusion criteria, and after the second screening, an additional 38 articles were excluded based on the criteria. In the first screening, 16 of the 382 articles were evaluated by a third reviewer. Subsequently, in the second screening, 15 of the 43 articles were also assessed by a third reviewer. We finally included five full-text articles [20–24] in this review (Figure 1).

*3.2. Study Characteristics*

3.2.1. Overview of Included Studies

The characteristics of the five included studies, all randomised controlled trials, are summarized in Table 1. The studies were published in the Netherlands [20], the United States [21], Turkey [22], Hong Kong [23], and Saudi Arabia [24], with the oldest article published in 2004 [21]. The lowest number of participants was 41 [22], the highest was 261 [23], and the mean age of the eligible patients or survivors of childhood cancer ranged from 7.9 [21] to 13.2 [20] years. Three studies targeted acute lymphoblastic leukaemia [21,22,24], and two studies targeted paediatric cancers (haematological malignancies, brain tumours, neurological tumours, solid tumours) [20,23]. Three studies targeted children undergoing cancer treatment [20,21,24] and three studies children after treatment [20,22,23]. Members of the studied group consisted of patients in two studies [20,23] and patients and their family members in three studies [21,22,24].

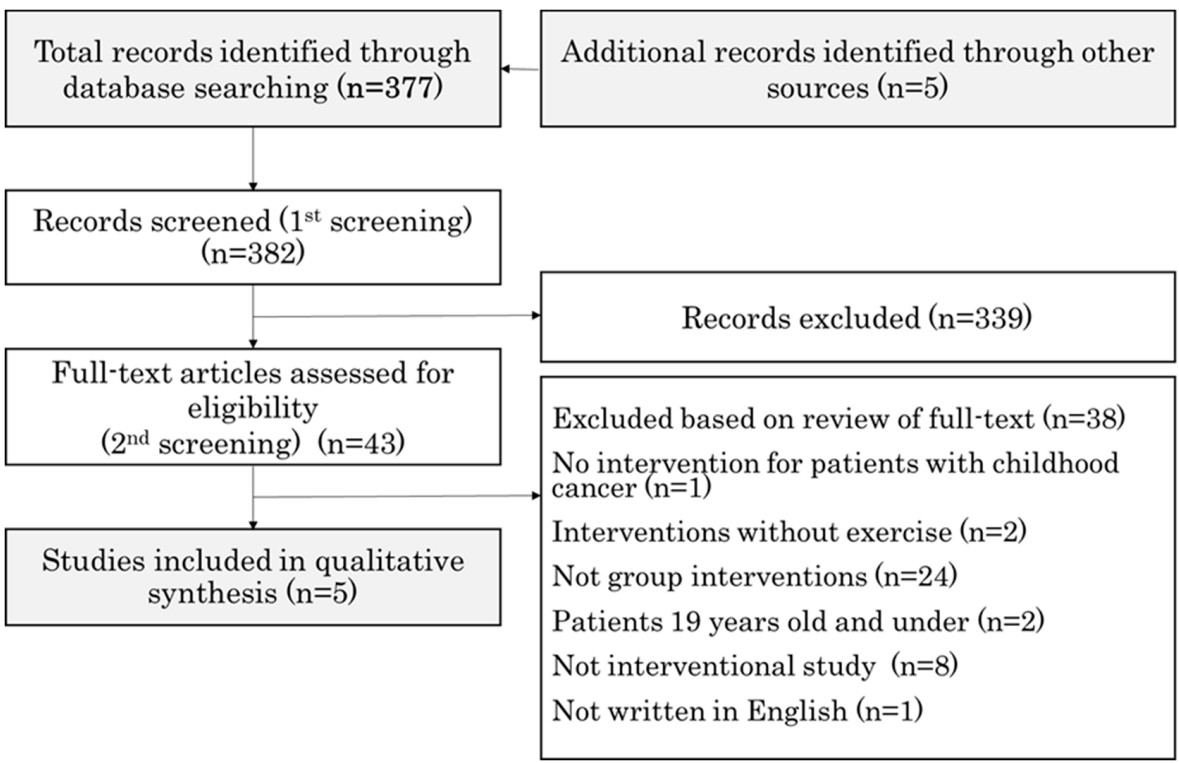

**Figure 1.** Search strategy flow diagram.

3.2.2. QOL

Four of the five studies assessed QOL, and all four studies used the PedsQL (Pediatric Quality of Life Inventory™) as an outcome measure of QOL [20–23]. The Dutch self-reported version of the PedsQL™ Generic Core Scales [20], PedsQL version 3.0 [21], PedsQL versions 3.0 and 4.0 [22], and the Chinese version [23] were used in these studies. The PedsQL is a tool for assessing QOL in paediatric patients [25]. The original PedsQL consists of four multi-item subscales: physical functioning (eight items), emotional functioning (five items), social functioning (five items), and school functioning (five items) [25]. The total QOL score is calculated based on the PedsQL manual and on a scale of 0–100, with higher scores reflecting a better QOL [25]. There are different versions of the PedsQL suitable for each age group and various languages [20–23,25]. PedsQL has been shown to be a reliable and valid assessment method in paediatric patients [25]. None of the four studies using the PedsQL in the present analysis indicated whether the children reliably completed the questionnaires [20–23].

Three of these four studies showed improvement in QOL [20,22,23]. One study found significant associations between psychosocial variables (depressive symptoms, athletic competence, global self-worth, and behavioural problems) and QOL, although no significant associations with QOL were found for physical variables and fatigue [20]. In one study, a 3-month exercise program resulted in decidedly significant increases in both the intervention and control groups at the final measurement as compared with the initial scores in the subscales of pain and hurt, nausea, and procedure-related anxiety on the PedsQL 3.0 Cancer Module Children's Form [22]. In another study, the adventure-based training that motivated survivors to maintain their physical activity could effectively and progressively reduce cancer-related fatigue, which may have improved QOL due to reduced fatigue levels [23].

**Table 1.** Characteristics of the 5 included studies.

| Study | Number of Participants | Patients (Age/Diagnosis) | Intervention (Type of Exercise, Duration, and Follow-Up) | Group Intervention Element | Outcomes | Main Findings |
|-------|------------------------|--------------------------|----------------------------------------------------------|----------------------------|----------|---------------|
| Braam et al., 2018 | 174 Completed: 53 | Intervention group (M = 13.4; SD = 3.1 /ALL, AML, HL, non-HL, CML, Burkitt, CNS, Brain tumour, Solid tumour) Control group (M = 13.1; SD = 3.1 /same as above) | Psychosocial training Physical exercise training: • Included both aerobic and weight-bearing exercises performed • Training was performed at an HR peak of 66–77% in the 1st month, 77–90% in 2nd month, 90–100% in 3rd month Home-based program: • A number of weight-bearing exercises at a high intensity level Duration: 3 months Follow-up: 4 months and 12 months | Tennis lesson, which was provided to a small group of age-matched study participants | • QOL (Dutch self-reported version of the PedsQL™ Generic Core Scales) • Cardiorespiratory fitness • Muscle strength • Physical activity • Body composition • Fatigue • Athletic competence and global self-worth • Behavioural problems • Depressive symptoms | • Intervention group showed larger improvements in lower-body muscle strength at 12 months when compared to control group • Bone mineral density and QOL improved significantly over time in both the intervention and control groups • In the intervention group, both upper and lower muscle strengths improved significantly over time |
| Marchese et al., 2004 | 33 Completed: 28 | Intervention group (5.1–15.8/ALL) Control group (4.3–10.6 /same as above) | • Received 5 sessions of physical therapy and participated in a home exercise program • Intervention program was based on activities the child and family enjoyed, a review of the pre-test physical assessment, and observation of the child's mobility, walking, and running Home exercise program: • Bilateral ankle dorsiflexion stretching • Bilateral lower extremity strengthening • Aerobic fitness (such as walking, biking, or swimming) Duration: 3 months Follow-up: 4 months | The program was based on activities the child and family enjoyed | • QOL (PedsQL version 3.0) • Muscle strength • Range of motion • Endurance • Functional mobility | • Significant increases in ankle dorsiflexion active range of motion in the intervention group • Knee extension strength increased significantly for the children in the intervention group • No significant differences between groups for ankle dorsiflexion strength, timed up and down stairs test, 9 min run-walk, and QOL score from pre- to post-test |

**Table 1.** *Cont.*

| Study | Number of Participants | Patients (Age/Diagnosis) | Intervention (Type of Exercise, Duration, and Follow-Up) | Group Intervention Element | Outcomes | Main Findings |
|---|---|---|---|---|---|---|
| Tanir et al., 2013 | 41 Completed: 40 | Trial group (M = 10.2; SD = 1.5 /ALL)<br><br>Control group (M = 10.7; SD = 1.5 /same as above) | • Active range of motion<br>• Leg exercises for strengthening the muscles<br>• Aerobic exercises with one of the following suggested exercises: Step-dancing to the rhythm of music, Jumping rope, Riding a bicycle, Running at a slow pace, in rhythm, Walking quickly<br><br>Duration: 3 months<br>Follow-up: 3 months | One of each child's parents was admitted into the session | • QOL (PedsQL versions 3.0 and 4.0)<br>• Physical parameters (9 min walk test, timed up and down stairs test, timed up and go test, back and leg dynamometry, and goniometer tests) | • Significant increases in the QOL subscales of pain and hurt, nausea, and procedure-related anxiety in both the trial and control groups<br>• Mean scores for the distance walked in the 9 min walk test, up and down stairs climbing times, up and go times, leg muscle strength measurement, and haemoglobin and haematocrit levels showed more improvement in the trial group compared to the control group |
| Li et al., 2018 | 261 Completed: 222 | Experimental group (M = 12.8; SD = 1.9 /Leukaemia, Lymphoma, Brain tumour, Bone tumour, Neuroblastoma)<br><br>Control group (M = 12.5; SD = 2.6 /same as above) | • Adventure-based training at a campsite<br>• Activities included ice-breaking and teambuilding games, shuttle runs, rock climbing, high- and low-level rope courses and descending<br><br>Duration: 6 months<br>Follow-up: 6 months and 12 months | The program was implemented in small groups, with a maximum of 12 participants | • QOL (Chinese version of the PedsQL)<br>• Cancer-related fatigue<br>• Physical activity levels<br>• Self-efficacy | • Participants in the experimental group reported lower levels of cancer-related fatigue, higher levels of physical activity and self-efficacy, and better QOL than those in the control group |
| Masoud et al., 2023 | 104 Completed: 45 | Intervention group (M = 9.0; SD = 2.3 /High risk ALL: 17, Standard risk ALL: 5)<br><br>Control group (M = 9.0; SD = 2.5 /High risk ALL: 15, Standard risk ALL: 8) | • Used exergaming<br>• Children can choose from the 23 applicable Wii games<br>• No limit to how many games the participant played as long as it was within the 60 min session, moderate intensity (50–70% increase of pred. HR max), and twice a week for three 3-week periods<br><br>Duration: 3 weeks<br>Follow-up: 5 weeks | Participants could play with one family member or the researcher | • Cancer-related fatigue<br>• Functional capacity/endurance<br>• Physical activity | • Significant reduction in cancer-related fatigue and significant improvement in physical activity and functional capacity/endurance were seen in the intervention group compared to the control group |

ALL, acute lymphoblastic leukaemia; AML, acute myeloid leukaemia; CML, chronic myeloid leukaemia; CNS, central nervous system; HL, Hodgkin lymphoma; HR, heart rate; M, mean; PedsQL, Pediatric Quality of Life Inventory™; QOL, quality of life; SD, standard deviation.

One study showed no improvement in QOL due to the majority of the participating children and parents reporting that they rarely or never had problems with the items on the PedsQL [21].

### 3.2.3. Physical Parameters

All five included studies used physical parameters as one of the outcome measures, with all studies showing improvement in these parameters, and three also showing improvement in QOL [20,22,23].

One of the three studies used upper limb muscle strength (shoulder, elbow, and grip strength), lower limb muscle strength (highest of hip, knee, and ankle dorsiflexion), peak oxygen uptake, and an Actical accelerometer as outcome measures for physical parameters. Consequently, significant improvement was found in lower limb strength at 12 months in the intervention group compared to the control group, and both upper and lower limb muscle strengths increased significantly over time in the intervention group [20].

In another study, the 9-Minute Walk Test, Timed Up and Down Stairs Test, Timed Up and Go Test, back and leg dynamometry, and goniometer tests were used as outcome measures for physical parameters. Consequently, measurements of mean distance walked in the 9-Minute Walk Test, up and down stairs climbing times, up and go times, and leg muscle strength showed greater improvement in the interventional group versus the control group counterparts [22].

In the remaining study, the Chinese University of Hong Kong Physical Activity Rating for Children and Youth was used as an outcome measure for physical parameters, and participants in the experimental group reported a higher physical activity level than those in the control group. This study also reported that adventure-based training enhanced self-efficacy and promoted the adoption and maintenance of regular physical activity among childhood cancer survivors [23].

One of the two studies finding no improvement in QOL used knee extension strength, ankle dorsiflexion strength and active range of motion, the Timed Up and Down Stairs Test, and the 9-Minute Run-Walk Test as outcome measures for physical parameters. As a result, the knee extension strength increased significantly in the intervention group, as did the active ankle dorsiflexion range of motion, but these improvement did not impact endurance [21].

The other study did not use QOL as an outcome measure but rather the 6-Min Walk Test and Godin-Shepard Leisure-Time Physical Activity Questionnaire as outcome measures for physical parameters. As a result, physical activity level and functional capacity/endurance increased significantly in the intervention group compared to the control group [24].

### 3.3. Risk of Bias in Studies

The risk of bias for each study and that for each domain are summarized in Figure 2. In the judgment process, 11 of the 25 domains (5 domains × 5 studies) were evaluated by a third reviewer. Most articles showed a low risk of bias in terms of missing outcome data [20–22,24]. All studies showed a high risk of bias with regard to the selection of the reported result because multiple methods were used to analyse the data [20–24]. Most of the studies showed a high risk of bias in terms of deviations from the intended interventions due to inadequate blinding and an inappropriate analysis to estimate the effect of assignment to intervention [20–22,24]. Most of the studies also showed a high risk of bias relating to the measurement of the outcome due to the inadequate blinding of the outcome assessors [21–24].

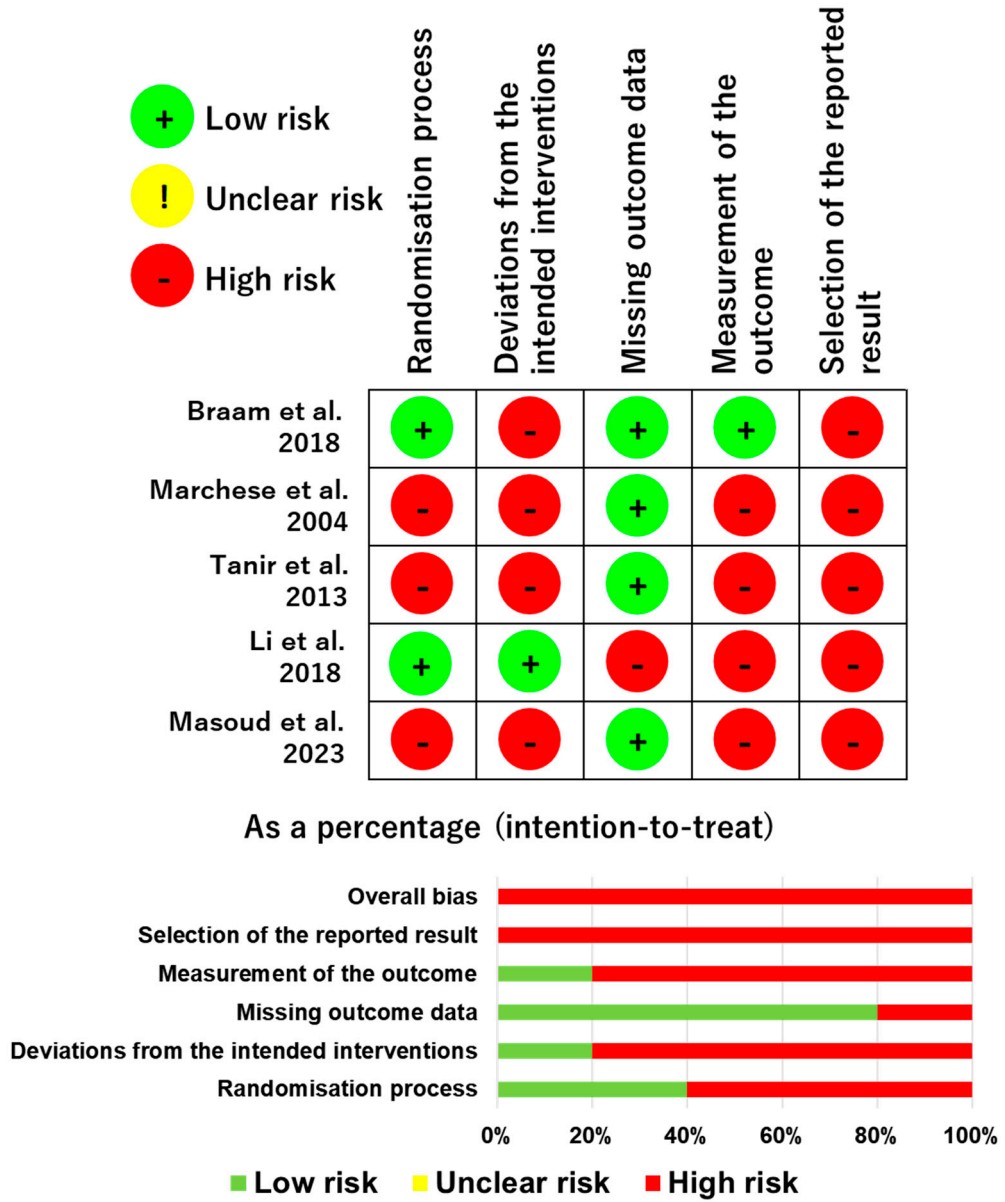

**Figure 2.** The risk of bias for each study and for each domain.

## 4. Discussion

This systematic review was performed to clarify the effect of a group exercise intervention on QOL and physical parameters in patients or survivors of childhood cancer. In three of the five studies analysed, both QOL and physical parameters improved after group exercise training, whereas in the other two studies, only physical parameters improved. The key findings of the present systematic review were that group exercise intervention supports the improvement of QOL in children with cancer and that it can be a beneficial way to enhance adherence to exercise training.

### 4.1. Group Exercise Intervention and QOL

A previous study showed that group-based exercise improved motor performance, overall physical activity level, and emotional well-being as a subscale of QOL in the childhood cancer population [17]. In our review, three of the five studies included showed improvements both in physical parameters such as muscle strength, endurance/cardiorespiratory fitness, and physical activity level and in QOL. It has also been reported that group-based exercise improved QOL in regard to physical and psychological symptoms. Thus, group

exercise interventions support psychological and physical effects and lead to improvement in QOL in terms of both physical and psychological symptoms. However, none of the studies included examined the difference in effect between individual and group interventions, so further research is needed to clarify the superiority of group exercise.

### 4.2. Group Exercise Intervention and Physical Parameters

Another previous study showed that the rate of exercise completion was low because it is sometimes hard to keep childhood cancer patients motivated in the training program for a one-on-one intervention [13]. Fun during exercising and participating in a social context with family and friends could motivate childhood cancer patients to participate in exercise training [26,27]. Thus, family and peer support is necessary to help these patients maintain motivation and complete exercise training. In addition, adherence to an exercise program has been shown to influence outcomes, so completion of an exercise program is important for improving physical parameters [14].

In our review, all five studies showed improvement in muscle strength, endurance/cardiorespiratory fitness, and physical activity level as physical parameters. A small number of participants were unable to complete the training due to a lack of motivation. One of five studies found that lack of motivation caused dropping out of training in 1 of 30 (3.3%) participants assigned to the intervention group [20]. Another study found the reason for dropping out of 14 of 117 (12%) participants assigned to the intervention group to be a lack of interest in participating [23]. In the other three studies, the reason for dropping out was either unclear or not related to lack of motivation. Thus, group exercise intervention contributes to maintaining motivation, and indeed, all five included studies suggested that many participants were able to complete the group exercise intervention. In each study, 22 of 30 (73.3%) [20], all 13 (100.0%) [21], 19 of 20 (95.0%) [22], 103 of 117 (88.0%) [23], and 22 of 23 (95.7%) [24] participants assigned to the intervention groups completed the exercise program. Thus, it is possible that the improvement in physical parameters in all five studies was due to the group exercise intervention preventing participants from losing motivation and completing the intervention program. However, none of the studies examined the impact of group interventions on participant motivation, so more studies are needed to investigate this relationship.

One previous study reported that family and peers act both as barriers to and as facilitators of participation in physical activity among childhood cancer survivors when the survivors perceive gaps in physical capacity and capability compared with their peers, parents do not have the time, siblings need parental attention, and parents regard physical activity as unimportant [27]. Nevertheless, our review found that many participants were able to complete the group exercise intervention with family or peer support. This finding supports the effectiveness of group intervention.

### 4.3. Clinical Implications

This systematic review showed the effects of a group exercise intervention on QOL and physical parameters in patients or survivors of childhood cancer. In three of the five studies analysed, both QOL and physical parameters improved after group exercise training, whereas in the other two studies, only physical parameters improved. Therefore, group exercise intervention supports the improvement of QOL and physical parameters in children with cancer and beneficially enhances adherence to exercise training. However, this systematic review includes some limitations, and it is necessary to carefully examine the intervention methods used. Moreover, the results of our systematic review suggest that to encourage children to participate in exercise, it is important to form groups of the same age and physical ability level and to provide detailed explanations to help parents understand the importance of exercise because the parents feel reluctance and fear towards exercising.

*4.4. Limitations*

The present review has several limitations. First, we used only PubMed as the search database. Because PubMed is a database specializing in the medical field, we thought we could obtain an adequate number of articles. However, the number of studies and samples reviewed was limited. Second, meta-analyses could not be performed because we could not find an adequate number of appropriate articles. Third, all studies showed a high risk of bias in terms of the selection of the reported result, and most studies showed a high risk of bias in terms of deviations from the intended interventions and measurement of the outcome. The domain of selection of the reported results showed problems related to data not being analysed according to a pre-specified plan and results not being selected from multiple eligible outcome measurements within the outcome domain and multiple eligible analyses. Problems in the domain of deviations from intended interventions were that participants, carers, and intervention deliverers were aware of intervention groups during the trial, and the analysis did not accurately estimate the intervention effect. The domain of measurement of the outcome included problems of outcome measurements not being appropriate, measurement or ascertainment of outcome differing between intervention groups, outcome assessors being aware of the intervention that study participants received, and outcome evaluations being influenced by knowledge of the intervention received. Therefore, the interpretation of the results should be closely scrutinized. In addition, further high-quality studies (e.g., those that perform analyses according to a pre-specified plan; blind participants, caregivers, and intervention providers; and evaluate outcomes rigorously) are necessary to prevent such risks of bias in future work. Fourth, the article by Braam et al. [20] among the studies included in this systematic review examined the effects of a combination of psychosocial and exercise intervention, so the improvement in QOL may not be brought about by group exercise interventions alone. Finally, none of the studies included compared individual and group interventions.

**5. Conclusions**

The group exercise intervention supports improvements in QOL and physical parameters in children with cancer and beneficially enhances adherence to exercise training. However, this systematic review highlighted some limitations of the reviewed studies, and it is necessary to carefully examine the intervention methods used. We suggest that to encourage children to participate in exercise, it is important to form groups of the same age and physical ability level and to provide detailed explanations to help parents understand the importance of exercises.

**Supplementary Materials:** The following supporting information can be downloaded at: https://www.mdpi.com/article/10.3390/curroncol31020077/s1, Table S1: Terms used in the search performed on 27 September 2023.

**Author Contributions:** Conceptualization, Y.D., M.O., K.I. and K.P.I.; methodology, Y.D., M.O., K.I. and K.P.I.; writing—original draft preparation, Y.D.; writing—review and editing, Y.D., M.O., K.I., K.P.I. and J.I.; supervision, K.P.I. All authors have read and agreed to the published version of the manuscript.

**Funding:** This research received no external funding.

**Acknowledgments:** We acknowledge the support of and encouragement from Ayami Osumi, Ayano Makihara, Ryo Yoshihara, Ikko Kubo, Asami Ogura, Yuji Kanejima, Masashi Kanai, Masahiro Kitamura, and Shinichi Shimada, Department of Public Health, School of Health Sciences, Kobe University.

**Conflicts of Interest:** The authors declare no conflicts of interest.

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
