# Peer review of "Effects of Group Exercise Intervention on Quality of Life and Physical Parameters in Patients with Childhood Cancer: A Systematic Review"

_curroncol, doi:10.3390/curroncol31020077_

Round 1

Reviewer 1 Report

Comments and Suggestions for Authors

See attached comments.

Comments on the Quality of English Language

See my review for examples of awkward or unclear writing. The article could benefit from your language editors.

Author Response

Response to Reviewer #1

We thank you for critically reading the manuscript and for providing us with meaningful comments and helpful suggestions. In accordance with your comments, we revised the manuscript as necessary. We are pleased that the revised manuscript has been improved by your great suggestions. Revisions are shown in red font color in the manuscript and as described below. Our responses to your comments are as follows:

Review of Effects of Group Exercise (Current Oncology, 1-22-24).

The review is on an interesting topic and reveals how little is known about the effects of exercise on young cancer patients. The review also reveals the differences in exercise approaches in various studies, making it even more difficult to make recommendations for young cancer patients. The authors indicate some encouraging results from an eventual review of 5 studies filtered down from 382 studies found in their search. The review merits consideration for publication, but it needs major improvement. I list here what could be done to improve the reporting of the review.

  1. The title of the article says: The effects of group exercise intervention on QOL…” but the review also looks at improvement in physical measures. Should this be added to the title?

Response: Thank you very much for your important suggestions. We agree. Per your suggestions, we revised the title.

Revisions

Title

Effects of Group Exercise Intervention on Quality of Life and Physical Parameters in Patients with Childhood Cancer: A Systematic Review

  1. Selection Process: provide the degree of agreement between the two reviewers and how many were subjected to discussion with the third reviewer.

Response: Thank you very much for your important suggestions. We agree and tried to calculate kappa coefficient. However, we could not calculate it because the results of the initial evaluation before the third-party evaluation could not be restored on the Rayyan software system. Thus, we described the number of papers discussed with a third reviewer in the Results sections.

Revision

  1. 3, L. 37-39

In the first screening, 16 of the 382 articles were evaluated by a third reviewer. Subsequently, in the second screening, 15 of the 43 articles were also assessed by a third reviewer.

  1. Risk of Bias Assessment: Need more details on the tool used to assess the risks in five domains and their definitions with some examples. I do not understand what is meant by bias in the randomization process, deviations from intended interventions, missing outcome data (could this cause problems with randomization), measurement of outcomes (does this refer to reliability/validity of outcome measures?), or selection of reported results (any issues with adequacy/appropriateness of statistical procedures used?). I do not get a picture of what was wrong in these studies about these 5 domains. Again, provide the degree of agreement between the two reviewers and how many were subjected to discussion with the third reviewer. The authors categorize studies as high, low, or unclear risk on the 5 domains (in Figure 2 they use the term “come concerns” – I think the descriptions “unclear” and “some concern” have different meanings). Also, provide more details on how the reviewers decided (criteria for classification) what category to assign to the 5 studies. How much agreement was there on this classification?

Response: Thank you very much for your important comments. We agree. First, we added more specific detail on the Risk of Bias Assessment in the Methods sections. Second, we tried to calculate kappa coefficient but could not as explained in our response to Comment 2 above. Thus, we described the number of papers discussed with a third reviewer in the Results sections. Finally, we revised the terms in Figure 2.

Revisions

  1. 3, L. 10-31

Risk of bias was evaluated for the following five domains: the randomisation process, deviations from the intended interventions, missing outcome data, measurement of the outcome, and selection of the reported result. The domain of the randomisation process was evaluated based on the assessment of random sequence generation, concealment of allocation sequence, and baseline imbalance and analyses that adjust for baseline imbalances [19]. The domain of deviations from intended interventions was evaluated on whether participants, carers, and people delivering the interventions were unaware of intervention groups during the trial and whether an appropriate analysis was used to estimate the effect of assignment to intervention [19]. The domain of missing outcome data was evaluated on whether outcome data were available for all, or nearly all, randomised participants, and whether the result was not biased by missing outcome data and that missing outcome could not depend on their true value [19]. The domain of measurement of the outcome was evaluated on the appropriateness of outcome measurement, whether the measurement or ascertainment of the outcome did not differ between intervention groups, that the outcome assessors were aware of the intervention received by the study participants, and whether assessment of the outcome was influenced by knowledge of the intervention received [19]. The domain of selection of the reported results was evaluated on whether the data were analysed in accordance with a pre-specified plan and whether the result was selected from multiple eligible outcome measurements within the outcome domain and multiple eligible analyses [19]. These five domains were ultimately adjudicated by an algorithm to determine risk of bias and reported as “low risk”, “unclear risk”, or “high risk” [19].

  1. 8, L. 3-4

In the judgment process, 11 of the 25 domains (5 domains × 5 studies) were evaluated by a third reviewer.

Figure 2

  1. Table 1. Give sample sizes for the experimental and control groups. Age is provided as, for example, 13.1±3.1; the ± symbol does not mean anything unless you compute confidence intervals where standard errors are used. If 3.1 is SD, put (M = 13.1; SD = 3.1). Prepare a legend for Acronyms used and put it as a footnote under the table for easier reading. I had to search the article for what they meant. The measures section should come before Table 1 to understand Table 1.

Response: Thank you very much for your important comments. We agree. Per your comments, we revised the Table 1 and placement of the Results section.

Revisions

Table 1 and placement of the Results section

  1. The authors state: “Four of the five studies included used PedsQL (Pediatric Quality of Life Inventory™) as an outcome measure of QOL [20-23]. The Dutch self-reported version of the PedsQL™ Generic Core Scales [20], PedsQL version 3.0 [21], PedsQL 3.0 and 4.0 [22], and the Chinese version [23] were used in these studies.”

The sentence is unclear, there are 6 scales—when one examines Table 1, it seems only 2 measures were used: QOL and HRQOL. So, in Table 1, it appears that QOL refers to any of the above measures. They also state HRQOL and Long-term HRQOL—it is not clear if they are the same measures or different measures—what does long-term mean? In Table 1 or anywhere else, the duration of intervention for the various studies is not indicated, an important consideration for discussion. It also appears that studies ended when the intervention ended and there was no follow-up to see how long the benefits lasted.

The authors talk about measurement bias, but there is no mention of whether the reviewed studies reported the reliability of the instruments found in their studies. This would be an important consideration in evaluating the quality of studies. It would be helpful to have more information on what these instruments measure, their subscales, and the reliability values found in the studies reviewed. The Figure on bias is nice, but hardly readable without magnifying it. Also, there are no details as to what sorts of biases were found in the reviewed studies, something that could be discussed in the Discussion section, outlining what future investigators need to consider in designing studies.

Response: Thank you very much for your important suggestions. We agree. First, the wording we used regarding QOL and HRQOL was misleading, so we revised the wording in Table 1 and the Results section. Second, we added more information and the reliability values and a reference about PedsQL in the Discussion section and References. Third, we revised Figure 2. Finally, we added more specific details on Risk of Bias in the Results and Discussion sections.

Revisions

Table 1 and Figure 2

References

  1. Varni, J.W.; Burwinkle, T.M.; Seid, M. The PedsQL as a pediatric patient-reported outcome: reliability and validity of the PedsQL Measurement Model in 25,000 children. Expert Rev Pharmacoecon Outcomes Res 2005, 5, 705-719, doi:10.1586/14737167.5.6.705.
  2. 4, L. 18-21

Four of the five studies assessed QOL, and all four studies used the PedsQL (Pediatric Quality of Life Inventory™) as an outcome measure of QOL [20-23]. The Dutch self-reported version of the PedsQL™ Generic Core Scales [20], PedsQL version 3.0 [21], PedsQL versions 3.0 and 4.0 [22], and the Chinese version [23] were used in these studies.

  1. 4, L. 22-25

One study found significant associations between psychosocial variables (depressive symptoms, athletic competence, global self-worth, and behavioural problems) and QOL, although no significant associations with QOL were found for physical variables and fatigue [20].

  1. 8, L. 3-10

In the judgment process, 11 of the 25 domains (5 domains × 5 studies) were evaluated by a third reviewer. Most articles showed a low risk of bias in terms of missing outcome data [20-22,24]. All studies showed a high risk of bias in regard to selection of the reported result because multiple methods were used to analyse the data [20-24]. Most of the studies showed a high risk of bias in terms of deviations from the intended interventions due to inadequate blinding and an inappropriate analysis to estimate the effect of assignment to intervention [20-22,24]. Most of the studies also showed a high risk of bias relating to measurement of the outcome due to inadequate blinding of the outcome assessors [21-24].

  1. 9, L. 19-27

As one method of evaluating QOL, all studies that assessed QOL used the PedsQL [20-23], which is a tool for assessing QOL in paediatric patients [25]. PedsQL consists of 4 multi-item subscales: physical functioning (8 items), emotional functioning (5 items), social functioning (5 items), and school functioning (5 items) [25]. The total QOL score is calculated based on the PedsQL manual and on a scale of 0–100, with higher scores reflecting a better QOL [25]. PedsQL was shown to be a reliable and valid assessment method in paediatric patients [25]. There are different versions of the PedsQL suitable for each age group and various languages [20-23,25]. Thus, PedsQL is recommended for the assessment of QOL in patients with childhood cancer.

  1. 10, L. 28-43

Third, all studies showed a high risk of bias in terms of the selection of the reported result, and most studies showed a high risk of bias in terms of deviations from the intended interventions and measurement of the outcome. The domain of selection of the reported results showed problems related to data not being analysed according to a pre-specified plan and results not being selected from multiple eligible outcome measurements within the outcome domain and multiple eligible analyses. Problems in the domain of deviations from intended interventions were that participants, carers, and intervention deliverers were aware of intervention groups during the trial, and the analysis did not accurately estimate the intervention effect. The domain of measurement of the outcome included problems of outcome measurements not being appropriate, measurement or ascertainment of outcome differing between intervention groups, outcome assessors being aware of the intervention that study participants received, and outcome evaluations being influenced by knowledge of the intervention received. Therefore, interpretation of the results should be closely scrutinized. Further high-quality studies to solve these risks of bias of group exercise intervention on QOL and physical parameters in patients with childhood cancer are required in future studies.

  1. Discussion could be much better—with all studies showing risk bias—the discussion needs to discuss the implications of such biases in designing future studies. Also, the inference “may improve QOL” or “could improve” is a weak one. This implies that the reviewing authors are not sure if the studies reporting improvement are valid. The reasons for such a vague conclusion should be clearly spelled out in a summary form in the conclusion section because it has implications for what medical practitioners should do—something that is not included in the Discussion section. In the conclusion section, the authors indicate that none of the studies compared “individual” with “group interventions”—this comment seems irrelevant if the purpose of the review was the look at only group interventions. Besides the sentence is awkwardly worded.

Response: Thank you very much for your important comments. We agree. Per your and another reviewer’s comments, first, we added more specific detail on Risk of Bias and implications for what medical practitioners should do in the Discussion section. Second, we revised wording using “may” in the Discussion section and added the reasons for our vague conclusion in the Conclusions section, which was entirely revised.

Revisions

  1. 9, L. 5-7

The key findings of the present systematic review were that group exercise intervention supports the improvement of QOL in children with cancer and that it may be a beneficial way to enhance adherence to exercise training.

  1. 9, L. 14-16

Thus, group exercise interventions support psychological and physical effects and lead to improvement in QOL in terms of both physical and psychological symptoms.

  1. 9, L. 33-34

Thus, family and peer support is necessary to help these patients maintain motivation and complete exercise training.

  1. 9, L. 44-46

Thus, group exercise intervention contributes to maintaining motivation, and indeed, all five included studies suggested that many participants were able to complete the group exercise intervention.

  1. 10, L. 10-22

4.3. Clinical implications

This systematic review showed the effects of a group exercise intervention on QOL and physical parameters in patients or survivors of childhood cancer. In three of the five studies analysed, both QOL and physical parameters improved after group exercise training, whereas in the other two studies, only physical parameters improved. Therefore, group exercise intervention supports improvement of QOL and physical parameters in children with cancer and beneficially enhances adherence to exercise training. However, this systematic review includes some limitations, and it is necessary to carefully examine the intervention methods used. Moreover, the results of our systematic review suggest that to encourage children to participate in exercise, it is important to form groups of the same age and physical ability level and to provide detailed explanations to help parents understand the importance of exercise because the parents feel reluctance and fear towards exercising.

  1. 10, L. 28-43: Please see Revisions to Comment 5.
  2. 10, L. 49–P. 11, L. 5

The group exercise intervention supports improvement of QOL and physical parameters in children with cancer and beneficially enhances adherence to exercise training. However, this systematic review highlighted some limitations of the reviewed studies, and it is necessary to carefully examine the intervention methods used. We suggest that to encourage children to participate in exercise, it is important to form groups of the same age and physical ability level and to provide detailed explanations to help parents understand the importance of exercise.

  1. Writing in some places can be improved for grammar and syntax. For example, “In answer to this problem,” (In an answer) “Therefore, the purpose of this study was to clarify the relationship between group exercise training and QOL or physical parameters in patients and survivors of childhood cancer via a thorough systematic review.” (instead of “or,” use “and” since you look at both and add say “review of published studies”) “Each of five domains of the randomization process” (perhaps say: Risk bias was evaluated for the following five domains: randomization process…). The way it is stated it can be interpreted to mean you are going to describe 5 domains of the randomization process.

Response: Thank you very much for your important comments. Per your comments, we improved grammar and syntax of problem sentences in the Introduction and Methods sections.

Revisions

  1. 2, L. 15-18

As one answer, exercise interventions that include family/peer interaction and a combined program of exercise and play have been reported to partially contribute to improvements in participation rate in exercise intervention, physical and psychological aspects, and QOL in patients and survivors of childhood cancer [15-17].

  1. 2, L. 20-22

Therefore, the purpose of this study was to clarify the relationship between group exercise training and QOL and physical parameters in patients and survivors of childhood cancer via a systematic review of published studies.

  1. 3, L. 10-31: Please see Revisions to comment 3.

Reviewer 2 Report

Comments and Suggestions for Authors

Dear Authors,

The necessity of your revision of the literature is well justified, and you used a correct methodology, the presentation of the results and the table with the main results of the selected studies are clear. Discussion and conclusion a in correspondence with your results.

But we do not understand for witch reason you limited you research to one database and only on studies from 2020 to 2023. For the database we imagine that you can justify because it is a very good database oriented on medical aspect, and with selected journal that give guarantees of publication quality. But for the limit to 2020 until 2023, we do not understand why! In a quick and superficial research, we find article of 2017 that may have be part of your inclusion criteria’s, then it is difficult understand why you limited to from 2020, with the eventual risk to lose some studies of interest for your revision objectives. Please include a justification in your methodology to explain why you limited to only one database, and why from 2020 and not earlier.

Wishes of success

Author Response

Response to Reviewer #2

We thank you for critically reading the manuscript and for providing us with meaningful comments and helpful suggestions. In accordance with your comments, we revised the manuscript as necessary. We are pleased that the revised manuscript has been improved by your great suggestions. Revisions are shown in red font color in the manuscript and as described below. Our responses to your comments are as follows:

Comments and Suggestions for Authors

Dear Authors,

The necessity of your revision of the literature is well justified, and you used a correct methodology, the presentation of the results and the table with the main results of the selected studies are clear. Discussion and conclusion a in correspondence with your results.

But we do not understand for witch reason you limited you research to one database and only on studies from 2020 to 2023. For the database we imagine that you can justify because it is a very good database oriented on medical aspect, and with selected journal that give guarantees of publication quality. But for the limit to 2020 until 2023, we do not understand why! In a quick and superficial research, we find article of 2017 that may have be part of your inclusion criteria’s, then it is difficult understand why you limited to from 2020, with the eventual risk to lose some studies of interest for your revision objectives. Please include a justification in your methodology to explain why you limited to only one database, and why from 2020 and not earlier.

Wishes of success

Response: Thank you very much for your important comments. We added the reason that we used the PubMed database in the Limitations portion of the Discussion section. As stated in the manuscript, we set the search period from January 2000 to September 2023, and we thought that there would be no problem in obtaining an adequate number of articles for review over this period.

Revisions

  1. 10, L. 24-27

First, we used only PubMed as the search database. Because PubMed is a database specializing in the medical field, we thought we could obtain an adequate number of articles. However, the number of studies and samples reviewed was limited.

Reviewer 3 Report

Comments and Suggestions for Authors

The Authors present an interesting systematic review on : " Effects of group exercise intervention on quality of life in patients with childhood cancer" . The topic is very important and should modify the present behavior to limit the physical activity of children with cancer.

I appreciated the modality used to perform this systematic review and the limitations reported in their final analysis.

Nevertheless the Authors should make the effort based on their literature analysis to express their idea from this analysis suggesting an action for facilitating and improving the participation of children to a physical exercise since the diagnosis overcoming parents' reluctance and fears first and foremost.

I don't like to read in the Conclusions " further research is needed to...." I need to read that an encouraging take home message is left to the readers.

Author Response

Response to Reviewer #3

We thank you for critically reading the manuscript and for providing us with meaningful comments and helpful suggestions. In accordance with your comments, we revised the manuscript as necessary. We are pleased that the revised manuscript has been improved by your great suggestions. Revisions are shown in red font color in the manuscript and as described below. Our responses to your comments are as follows:

Comments and Suggestions for Authors

The Authors present an interesting systematic review on: "Effects of group exercise intervention on quality of life in patients with childhood cancer". The topic is very important and should modify the present behavior to limit the physical activity of children with cancer.

I appreciated the modality used to perform this systematic review and the limitations reported in their final analysis.

Nevertheless the Authors should make the effort based on their literature analysis to express their idea from this analysis suggesting an action for facilitating and improving the participation of children to a physical exercise since the diagnosis overcoming parents' reluctance and fears first and foremost.

I don't like to read in the Conclusions "further research is needed to...." I need to read that an encouraging take home message is left to the readers.

Response: Thank you very much for your important comments. We agree. Per your and another reviewer’s comments, we added more specific detail of the idea of an action facilitating and improving the participation of children to a physical exercise in the Discussion sections. Moreover, we revised whole sentence in the Conclusions sections.

Revisions

  1. 11, L. 11-23

4.3. Clinical implications

This systematic review showed that the effect of a group exercise intervention on QOL and physical parameters in patients or survivors of childhood cancer. In three of the five studies analysed, both QOL and physical parameters improved after group exercise training, whereas in the other two studies, only physical parameters improved. Therefore, the group exercise intervention supports to improve QOL and physical parameters in children with cancer, and the beneficial way to enhance adherence to exercise training. However, this systematic review includes some limitations, and it is necessary to examine the intervention method carefully. Moreover, the results of our systematic review suggest that in order to encourage children to participate in exercise, it is important to form groups of the same age and physical ability level, and to provide detailed explanations to help parents understand the importance of exercise because of parents feel reluctance and fears towards exercising.

  1. 12, L. 1-7

The group exercise intervention supports to improve QOL and physical parameters in children with cancer, and the beneficial way to enhance adherence to exercise training. However, this systematic review includes some limitations, and it is necessary to examine the intervention method carefully. We suggest that in order to encourage children to participate in exercise, it is important to form groups of the same age and physical ability level, and to provide detailed explanations to help parents understand the importance of exercise.

Round 2

Reviewer 1 Report

Comments and Suggestions for Authors

Review of Current Oncology Revised Version2 (2/9/2024)

While the authors have addressed most of my comments, some of these comments were not addressed fully.

1. Results section:  As a result, a significant improvement in physical activity level compared to the control and a significant main effect of time × intervention interaction on functional capacity/endurance were shown [24].

In the above statement “a significant main effect of time × intervention interaction” is incorrect. Main effects are different from interaction effects, so the above sentence needs to be clarified. You need to explain what this interaction means—it is unclear how to interpret this result unless one refers to the original study.

2. “In Table 1 or anywhere else, the duration of intervention for the various studies is not noted, an important consideration for discussion. It also appears that studies ended when the intervention ended and there was no follow-up to see how long the benefits lasted.”

I did not see any comment about the duration of the studies or the follow-up in the revised version.

3. “The authors talk about measurement bias, but there is no mention of whether the reviewed studies reported the reliability of the instruments found in their studies.”

The reliability values of QOL obtained in the 5 studies need to be provided in the results section, not handled in the Discussion section in a general manner by providing other references by simply saying that the measures are reliable and valid. The question is: Did the children reliably complete the questionnaires in the studies reviewed?  If no information is available in the 5 reviewed studies—that should be noted. Also, the descriptions of the subscales, etc., included in section 4.1 should be in the Results section.

4.  Discussion could be much better—with all studies showing risk bias—the discussion needs to discuss the implications of such biases in designing future studies.

While they included clinical implications, there is no mention of what future studies need to do, especially given the risks of bias they noted—how to minimize the various risks. The risks are noted but I do not get an idea of what happened in two studies with respect to randomization and how the missing data were handled in these studies.

Author Response

Response to Reviewer #1

We thank you for critically reading the manuscript and for providing us with meaningful comments and helpful suggestions. In accordance with your comments, we revised the manuscript as necessary. We are pleased that the revised manuscript has been improved by your great suggestions. Revisions are shown in red font color in the manuscript and as described below. Our responses to your comments are as follows:

Comments and Suggestions for Authors

Review of Current Oncology Revised Version2 (2/9/2024)

While the authors have addressed most of my comments, some of these comments were not addressed fully.

  1. Results section: As a result, a significant improvement in physical activity level compared to the control and a significant main effect of time × intervention interaction on functional capacity/endurance were shown [24].

In the above statement “a significant main effect of time × intervention interaction” is incorrect. Main effects are different from interaction effects, so the above sentence needs to be clarified. You need to explain what this interaction means—it is unclear how to interpret this result unless one refers to the original study.

Response: Thank you very much for your important suggestions. We agree. The sentence was misleading, so we revised the sentence in the Results section and Table 1.

Revisions

Table 1

  1. 5, L. 45-47

As a result, physical activity level and functional capacity/endurance increased significantly in the intervention group compared to the control group [24].

  1. “In Table 1 or anywhere else, the duration of intervention for the various studies is not noted, an important consideration for discussion. It also appears that studies ended when the intervention ended and there was no follow-up to see how long the benefits lasted.”

I did not see any comment about the duration of the studies or the follow-up in the revised version.

Response: Thank you very much for your important suggestions. We agree. Per your suggestions, we added the duration of intervention and follow-up for the studies in Table 1.

Revisions

Table 1

  1. “The authors talk about measurement bias, but there is no mention of whether the reviewed studies reported the reliability of the instruments found in their studies.”

The reliability values of QOL obtained in the 5 studies need to be provided in the results section, not handled in the Discussion section in a general manner by providing other references by simply saying that the measures are reliable and valid. The question is: Did the children reliably complete the questionnaires in the studies reviewed? If no information is available in the 5 reviewed studies—that should be noted. Also, the descriptions of the subscales, etc., included in section 4.1 should be in the Results section.

Response: Thank you very much for your important comments. We agree. Per your comments, we added appropriate text in the Results section.

Revisions

  1. 4, L. 22-30

The PedsQL is a tool for assessing QOL in paediatric patients [25]. The original PedsQL consists of 4 multi-item subscales: physical functioning (8 items), emotional functioning (5 items), social functioning (5 items), and school functioning (5 items) [25]. The total QOL score is calculated based on the PedsQL manual and on a scale of 0–100, with higher scores reflecting a better QOL [25]. There are different versions of the PedsQL suitable for each age group and various languages [20-23,25]. PedsQL has been shown to be a reliable and valid assessment method in paediatric patients [25]. None of the four studies using the PedsQL in the present analysis indicated whether the children reliably completed the questionnaires [20-23].

  1. Discussion could be much better—with all studies showing risk bias—the discussion needs to discuss the implications of such biases in designing future studies.

While they included clinical implications, there is no mention of what future studies need to do, especially given the risks of bias they noted—how to minimize the various risks. The risks are noted but I do not get an idea of what happened in two studies with respect to randomization and how the missing data were handled in these studies.

Submission Date

11 January 2024

Date of this review

09 Feb 2024 15:14:32

Response: Thank you very much for your important comments. We agree. Per your comments, we added the way that studies should deal with the risks of bias we noted in the Discussion section. We could neither find the details on randomisation in any of the studies nor how missing data were handled in two of the studies.

Revisions

  1. 11, L. 34-37

In addition, further high-quality studies (e.g., those that analyse according to a pre-specified plan; blind participants, caregivers, and intervention providers; and evaluate outcomes rigorously) are necessary to prevent such risks of bias in future studies.
